# Predictors of discharge disposition and mortality following hospitalization with SARS-CoV-2 infection

Farha Ikramuddin[1]*, Tanya Melnik[2], Nicholas E. Ingraham[2], Nguyen Nguyen[1], Lianne Siegel[3], Michael G. Usher[4], Christopher J. Tignanelli[5], Leslie Morse[1]

1 Department of Rehabilitation Medicine, Division of PM&R, University of Minnesota, Minneapolis, MN, United States of America, 2 Department of Medicine, Division of Pulmonary and Critical Care, University of Minnesota, Minneapolis, MN, United States of America, 3 Division of Biostatistics, School of Public Health, University of Minnesota, Minneapolis, MN, United States of America, 4 Department of Medicine, Division of General Internal Medicine, University of Minnesota, Minneapolis, MN, United States of America, 5 Department of Surgery, University of Minnesota Division of Acute Care Surgery, Minneapolis, MN, United States of America

* ikram002@umn.edu

**Data Availability Statement:** All relevant data are within the paper and its Supporting Information files.

## Abstract

### Importance

The SARS-CoV-2 pandemic has overwhelmed hospital capacity, prioritizing the need to understand factors associated with type of discharge disposition.

### Objective

Characterization of disposition associated factors following SARS-CoV-2.

### Design

Retrospective study of SARS-CoV-2 positive patients from March 7th, 2020, to May 4th, 2022, requiring hospitalization.

### Setting

Midwest academic health-system.

### Participants

Patients above the age 18 years admitted with PCR + SARS-CoV-2.

### Intervention

None.

### Main outcomes

Discharge to home versus PAC (inpatient rehabilitation facility (IRF), skilled-nursing facility (SNF), long-term acute care (LTACH)), or died/hospice while hospitalized (DH).

**Funding:** Funding for this study was provided by National Institute of Health's National Center for advancing Translational Sciences Grant U01TR002062, however the funders had no role in study design, data collection and analysis, decision to publish, or preparation of the manuscript.

**Competing interests:** The authors have declared that no competing interests exist.

## Results

We identified 62,279 SARS-CoV-2 PCR+ patients; 6,248 required hospitalizations, of whom 4611(73.8%) were discharged home, 985 (15.8%) to PAC and 652 (10.4%) died in hospital (DH).

Patients discharged to PAC had a higher median age (75.7 years, IQR: 65.6–85.1) compared to those discharged home (57.0 years, IQR: 38.2–69.9), and had longer mean length of stay (LOS) 14.7 days, SD: 14.0) compared to discharge home (5.8 days, SD: 5.9).

Older age (RRR:1.04, 95% CI:1.041–1.055), and higher Elixhauser comorbidity index [EI] (RRR:1.19, 95% CI:1.168–1.218) were associated with higher rate of discharge to PAC versus home. Older age (RRR:1.069, 95% CI:1.060–1.077) and higher EI (RRR:1.09, 95% CI:1.071–1.126) were associated with more frequent DH versus home. Blacks, Asians, and Hispanics were less likely to be discharged to PAC (RRR, 0.64 CI 0.47–0.88), (RRR 0.48 CI 0.34–0.67) and (RRR 0.586 CI 0.352–0.975). Having alpha variant was associated with less frequent PAC discharge versus home (RRR 0.589 CI 0.444–780).

The relative risks for DH were lower with a higher platelet count 0.998 (CI 0.99–0.99) and albumin levels 0.342 (CI 0.26–0.45), and higher with increased CRP (RRR 1.006 CI 1.004–1.007) and D-Dimer (RRR 1.070 CI 1.039–1.101). Increased albumin had lower risk to PAC discharge (RRR 0.630 CI 0.497–0.798. An increase in D-Dimer (RRR1.033 CI 1.002–1.064) and CRP (RRR1.002 CI1.001–1.004) was associated with higher risk of PAC discharge. A breakthrough (BT) infection was associated with lower likelihood of DH and PAC.

## Conclusion

Older age, higher EI, CRP and D-Dimer are associated with PAC and DH discharges following hospitalization with COVID-19 infection. BT infection reduces the likelihood of being discharged to PAC and DH.

**Question:** What factors, both baseline and intrahospital, associate with the need for and type of post-acute discharge following hospitalization for SARS-CoV-2?

**Findings:** Analysis of this retrospective electronic health record cohort of SARS-CoV-2 patients (n = 6248) hospitalized from March 7th, 2020 –May 4th, 2022, revealed that being older, having higher baseline inflammatory markers, and having more comorbidities increases the likelihood of discharge to a PAC (post-acute care facilities). Discharge to PAC is associated with higher mortality based on state death certificate-based mortality rates. Discharges to PAC are mitigated for patients who develop COVID-19 post vaccination and who have the Alpha variant. Furthermore, Blacks and Asians are more likely to be discharged to home versus PAC.

*Meaning*: Our findings demonstrated that those admitted with older age and higher comorbidities have increased PAC needs. We find that vaccination is protective to prolonged LOS, ICU stay, number of days in the ICU and death during hospitalization. Patients discharged to PAC have a higher mortality rate on analysis of state death certificate-based mortality rate. Having the alpha variant or a breakthrough infection are more associated with a home discharge than PAC or to die in hospital respectively. Finally,

Blacks and Asians are more likely to be discharged home when controlling for other baseline factors, a finding that warrants further study. Of the baseline COVID-19 symptoms obtained from natural language processing, only dyspnea was found to impact disposition by reducing the association of PAC utilization.

## Introduction

As of May 22nd, 2022, 90 million individuals have tested positive for SARS-CoV-2 and more than one million deaths have resulted from SARS-COV-2 in the US alone [1,2]. When severe, SARS-COV-2 is associated with a prolonged hospitalization, increased ICU stay and the need for mechanical ventilation [3]. Within 3 months from the start of the pandemic, reports emerged that patients hospitalized with SARS CoV-2 infection were developing impaired physical function thus driving the need for inpatient post-acute care (PAC) facilities which include the inpatient rehabilitation facility(IRF), skilled nursing facilities (SNF) and long term acute care hospitals (LTACH) [4]. Extrapolating from the sepsis literature, functional deficits after infection stem from multiple factors including prolonged hospitalization, ICU stay, intubation, use of steroids, and other systemic complications [5]. As also observed in sepsis, critical illness myopathies, and debility following prolonged intubation, ICU stays, and hospitalization add to the burden of care thus making discharge to home from the hospital challenging [6–9]. Neurological complications also add to the physical burden and subsequent PAC needs in the SARS-CoV-2 patients population [10]. Cryptogenic stroke has been noted to be twice as common in SARS-COV-2 patients. Encephalopathy and peripheral neuropathies compound these impairments.[7] In one study, half of the hospitalized SARS-COV-2 patients were found to have severe impairments in function and performance in activities of daily living [4]. The long-term comorbidities and functional impairment may lead to need for PAC [11].

There are limited data on the outcomes of the inpatient SARS-COV-2 patients following discharge from the acute care hospitals. Given the functional decline seen in these patient population on hospitalization, rehabilitation and discharge disposition is an important pathway to quality of life and return to work. Historically, discharge disposition has been associated with functional status at the time of discharge from the hospital and functional status at discharge is strongly associated with readmission rate [12,13].

A previous study, conducted prior to the general availability of vaccination, analyzed clinical characteristics of SARS-CoV-2 patients admitted to the hospital. The authors observed higher morbidity and mortality in patients with older age, male gender, Hispanic ethnicity, and those with bacterial co-infections and chronic comorbidities [14]. However, there is paucity of studies addressing the predictors of discharge disposition following hospitalization and specifically the PAC needs of the COVID-19 population. In the early part of the pandemic, many factors impacted discharge disposition, including the continued need for isolation on discharge to home, lack of a support system, reduced access to outpatient dialysis (as many dialysis centers did not accept SARS-CoV-2 patients), local healthcare culture, and payor practices and the availability of rehabilitation facilities. As the pandemic evolved, these factors and specifically the rehabilitation needs of this patient population became more pronounced.

With the continued emergence of new and often highly contagious SARS-CoV2 variants, healthcare systems already strapped with reduced staffing continue to be challenged with waxing and waning hospitalization rates [15].

Vaccines have been found to decrease the transmission of breakthrough infections, 40–50% following a single dose and 70% after full vaccination in a studies from the UK [16] and the

Netherlands [17]. While vaccination is noted to reduce the rate of hospitalization [18], discharge disposition and rehabilitation needs based on variants and immunization status has not been studied.

Changes in health care policy with the goal to increase patient access, an aging population, coupled with challenging staffing ratios and burnout even preceding the pandemic have most likely contributed to the healthcare staffing shortages impacting inpatient and post-acute care. Recognition of factors associated with utilization of PAC following discharge from the hospitals is of interest to all members of the health care team. Further characterization of factors driving PAC needs requires consideration of new virus variants; the emergence of (breakthrough) infections after vaccination is also of immense interest. We report our analysis of a large database of SARS-COV-2 patients to assess baseline factors associated with discharge disposition from the acute care setting.

## Methods

### Data collection

Data were abstracted from electronic health records (EHR) reports from a single academic health care system composed of 12 US Midwest hospitals and 60 primary clinics across Minnesota.

The study was approved by all hospitals within the M Health Fairview system which includes ethical approval by the University of Minnesota institutional board. All patients have the option to opt out of research upon establishing care within the MHealth Fairview healthcare system. Data were pooled across different EHRs utilizing a unique patient identifier to account for health care encounters across systems. This study was approved by the University of Minnesota institutional review board (STUDY00001489) which provided a waiver of consent for this study and demographics since March 29th, 1997, and was approved by all hospitals within the M Health Fairview system. State death certificates were linked to the database and enabled accurate out-of-hospital death data for each patient.

### Participants

Patients testing PCR positive for the SARS-CoV-2 (SARS- CoV-2 group) during the period of March 7th, 2020, to May 4th, 2022, were included, (Fig 1). We also chose to include patients admitted with a primary diagnosis of influenza from January 6, 2011, to November 14, 2020, as a control group. We could not to include the influenza patients during the pandemic as that specific data was lacking after November 2020.

Both the SARS- CoV-2 group and the influenza group consisted of patients requiring hospital admission to one of the 12 hospitals included in the database and who had a specified discharge disposition.

Variables were selected *a priori* based on their reported association with SARS-COV-2 morbidity, mortality, known pathophysiology in previous literature. Variables included age, race, gender, BMI, hemoglobin, white blood cell count, platelet count, preadmission albumin, blood type, Rh positive, CRP, Tumor necrosis factor TNF, D Dimer, troponin, hospital length of stay LOS, admission to ICU, ICU LOS, Elixhauser comorbidity index (EI)[15] and comorbidities. The data also included those who received Remdesivir, dexamethasone, and tocilizumab.

We combined the analysis of patients who died in hospital with those who were placed into hospice. The died in the hospital/hospice end point is a dependent composite endpoint and will from here be referred to as 'died in hospital (DH)'. 'Mortality' was any death that occurred following discharge from the hospital/PAC outside the hospital/facilities based on state certificate data.

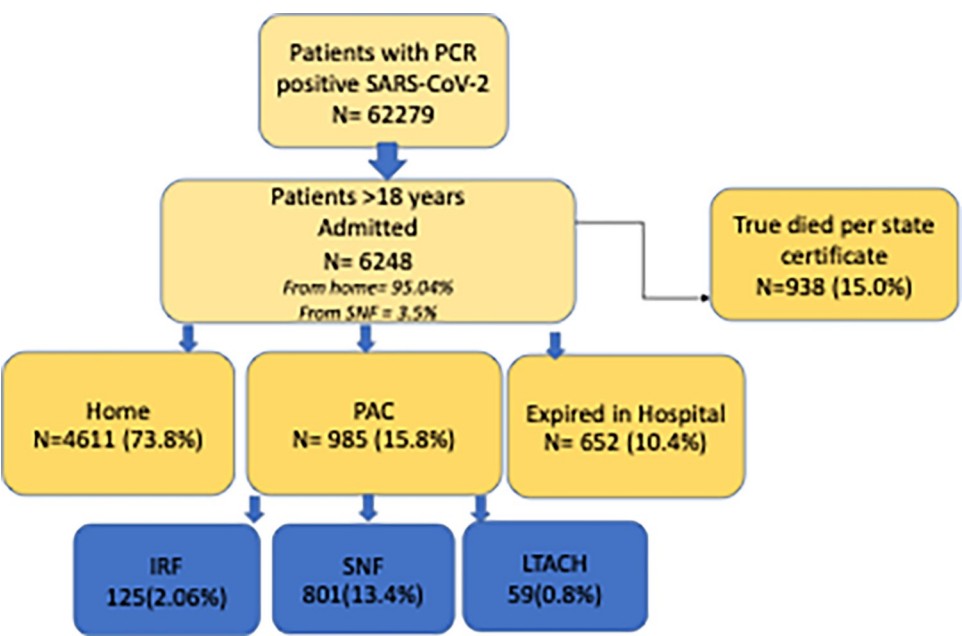

**Fig 1. Study flow chart.** Panel depicts participants agreeing to research selected from the electronic medical record. 62,279 PCR positive patients presented to the hospital systems, 6248 admitted to the hospital were analyzed for discharge disposition of PAC: Post-acute care, IRF: Inpatient rehabilitation unit, SNF: Skilled nursing home, LTACH: Long term care unit.

Additional outcomes included the need for extracorporeal membrane oxygenation ECMO.

We categorized discharge disposition to post-acute care (PAC) facilities as a group in addition to describing the discharge dispositions within the PAC group. The post-acute care (PAC) group included inpatient rehabilitation facility (IRF), skilled nursing facility (SNF), and long-term hospital (LTACH).

The variants were based on the timeline of the variants from the GISAID Database (https://covariants.org/per-country?region=World). The dates for the Epsilon variant were Sep 9th, 2020- Mar 8th, 2021, Alpha Variant; Mar 8th, 2021- Jun 14th, 2021, Delta Variant was Jun21st 2021- Dec13th 2021, and Omicron Variant was Dec 13th, 2021, to present [19]. Earlier variants included Epsilon and Robin1.

The main vaccines used in our population were Pfizer-BioNTech, Moderna Bivalent, and Johnson and Johnson's Janssen vaccines.

Patients were included in the database as vaccinated if they were vaccinated at least with one dose and developed COVID-19 after 2 weeks of being vaccinated. The timeline of two weeks was considered needed to develop immunity against COVID-19 infection.

## Statistical analysis

Continuous variables nonparametric variables were compared across the groups using a Mann Whitney test or Kruskal-Wallis test and expressed as median and interquartile range (IQR). Parametric continuous variables were reported as mean and standard deviation. Categorical variables were summarized using counts and percentages and compared across the groups using a Pearson chi-squared test or Fisher's exact test. Post hoc analysis to determine pairwise differences following the Kruskal Wallis test was carried out using the Dunn's test. Fisher's exact test was used to compare presenting symptoms across the discharge disposition. Two-sided p-values less than 0.05 were considered statistically significant.

The overall missingness rate of independent variables included in the primary analysis was 18.2%. The variables were selected based on a model *a priori* Missingness was addressed using multiple imputation by chained equations; a total of 10 imputed datasets were created.

A regression model was created to compare factors which were significantly associated with discharge to PAC versus home. Descriptive analysis of the patients admitted was completed to study the characteristics. Logistic regression was used for binary variables. Linear regression was used for continuous variables using the auxiliary variable that contained data for the minimum oxygen saturation for the first 24 hours. All analysis was conducted using Stata version 17.0 (Stata Corp, College Station, TX).

## Results

A total of 62,279 PCR positive SARS-CoV2 patients presented to the healthcare system between March 7[th], 2020, and May 4[th],2022. 6248 (10.0%) were admitted to the hospital with a documented discharge disposition. 95.04% of these patients were admitted from home through the emergency department (ED) or outpatient clinics, and 3.6% were from SNF. The remainder 1.46% had been transferred from hospitals outside the healthcare system.

On analysis of discharge disposition, 4611(73.8%) were discharged home, 985 (15.8%) to PAC and 652 (10.4%) died while being hospitalized (DH). The PAC group included 125 (2.06%) discharged to IRF, 801 (13.4%) to SNF and 59 (0.8%) to LTACH.

### Characteristics of the SARS-CoV-2 patient by discharge disposition: (Table 1)

Table 1 compares the baseline demographic and inpatient characteristics of SARS-COV-2 patients across discharge dispositions. The median age (IQR) of patients discharged to PAC was 75.7 years (65.6–85.1) compared to 57.0 years among those discharged home (38.2–69.9). Those in the DH group were the oldest 78.1 (66.9–85.6). Lower median albumin levels (IQR) on admission were observed in those discharged to PAC 2.7 (2.3–3.0) gm/dl and DH 2.5 (2.2–2.8) gm/dl compared to those discharged home 2.9 (2.3–3.0) gm/dl. Inflammatory biomarkers and markers of coagulation were analyzed; the normal ranges of these laboratory values are found in S1 Table. Higher median (IQR) CRP, and D-Dimer levels on admission were seen in patients DH: CRP 117.0 (63.5–176.0) mg/dl, D-Dimer 1.8 (1.1–3.7), and discharged to PAC: CRP 73 (32.0–129.0) mg/dl, and D-Dimer 1.3 (0.7–2.4), compared to those discharged to home: CRP 67.0 (29.0–119.0) mg/dl, D-Dimer 1.0 (0.6–1.7). Patients who were discharged to PAC and those who DH had a longer mean (SD) hospital length of stay LOS 14.7 (14.0) days and 13.3 (11.3) days compared to those discharged home 5.8 (5.9) days. More frequent ICU admission, longer ICU LOS, and more frequent ECMO utilization were observed in those in the DH group. Patients having PAC discharges had been admitted to the ICU more frequently than those discharged home (34.8% versus 18.1%, p<0.001). Additionally, patients who had been discharged to PAC had a longer mean ICU stay (6.0 days, SD: 13) compared to those who were discharged home (1.1 days, SD: 6.2, p<0.001). The mortality rate based on state death certificates was 4% for those discharged home, and 15.2% in those discharged to PAC (p<0.001). Those who had been discharged home had a lower median Elixhauser comorbidity index (4.0, IQR: 2.0–8.0), compared to those who were discharged to PAC (9.0, IQR: 7.0–12.0, p<0.001).

### Discharge disposition based on variants: (Table 2)

Variants were determined based on the timeline from the GISAID (https://covariants.org/per-country?region=World). The in-hospital mortality was the highest with the early variants at 16.8%, 10% in patients with alpha variant, 9.0% in the delta and 5.9% in omicron population. In our analysis, 55% of the Omicron patients were breakthrough (BT) infections, 28.6% in the

**Table 1. Demographics and characteristics of patients admitted with COVID-19 based on discharge disposition.** Categorical variables expressed as count and percentage, and continuous variables as median and IQR.

| | | Home | Post-Acute Care | Hospice/Expired | p-value |
|---|---|---|---|---|---|
| N | | 4,611 | 985 | N = 652 | |
| Age | | 57.0 (38.2–69.9) | 75.7 (65.6–85.1) | 78.1 (66.9–85.6) | <0.001 |
| Breakthrough | NO | 4,259 (92.4%) | 914 (92.8%) | 620 (95.1%) | 0.043 |
| | YES | 352 (7.6%) | 71 (7.2%) | 32 (4.9%) | |
| Race | White | 2,798 (63.1%) | 764 (83.1%) | 448 (72.6%) | <0.001 |
| | Black | 564 (12.7%) | 63 (6.9%) | 52 (8.4%) | |
| | Asian | 527 (11.9%) | 49 (5.3%) | 77 (12.5%) | |
| | Hispanic | 349 (7.9%) | 19 (2.1%) | 21 (3.4%) | |
| | Declined | 108 (2.4%) | 14 (1.5%) | 7 (1.1%) | |
| | Other | 85 (1.9%) | 10 (1.1%) | 12 (1.9%) | |
| Male | | 2,161 (46.9%) | 477 (48.4%) | 353 (54.1%) | 0.002 |
| BMI | | 29.9 (25.7–35.3) | 28.7 (24.2–33.7) | 28.0 (23.8–32.5) | <0.001 |
| Hemoglobin >15.7 g/dl | high | 141 (3.5%) | 23 (2.5%) | 20 (3.3%) | <0.001 |
| <11.7 g/dl | low | 1,438 (35.9%) | 419 (46.0%) | 284 (47.2%) | |
| 11.7–15.7 g/dl | normal | 2,430 (60.6%) | 468 (51.4%) | 298 (49.5%) | |
| WBC >11/ul | high | 496 (16.1%) | 157 (20.6%) | 111 (24.6%) | <0.001 |
| >4/ul | low | 232 (7.5%) | 38 (5.0%) | 27 (6.0%) | |
| 4-11/ul | normal | 2,350 (76.3%) | 566 (74.4%) | 314 (69.5%) | |
| Platelets >450/ul | high | 78 (2.0%) | 10 (1.1%) | 10 (1.7%) | <0.001 |
| <150/ul | low | 846 (21.8%) | 272 (29.7%) | 206 (34.3%) | |
| 150-450/ul | normal | 2,952 (76.2%) | 633 (69.2%) | 385 (64.1%) | |
| Albumin | | 2.9 (2.5–3.2) | 2.7 (2.3–3.0) | 2.5 (2.2–2.8) | <0.001 |
| Blood type | O | 1,141 (40.6%) | 267 (39.3%) | 183 (41.5%) | 0.95 |
| | A | 1,100 (39.2%) | 277 (40.7%) | 168 (38.1%) | |
| | B | 431 (15.3%) | 99 (14.6%) | 67 (15.2%) | |
| | AB | 136 (4.8%) | 37 (5.4%) | 23 (5.2%) | |
| Rh pos | | 2,526 (90.0%) | 600 (88.2%) | 388 (88.0%) | 0.24 |
| CRP | | 67.0 (29.0–119.0) | 73.0 (32.0–129.0) | 117.0 (63.5–176.0) | <0.001 |
| TNF | | 22.9 (18.5–32.5) | 25.0 (19.3–33.9) | 32.2 (21.7–41.8) | <0.001 |
| D-DIMER | | 1.0 (0.6–1.7) | 1.3 (0.7–2.4) | 1.8 (1.1–3.7) | <0.001 |
| Troponin | | 1.1 (13.0) | 0.6 (5.6) | 4.0 (56.5) | 0.057 |
| Inpatient LOS | | 5.8 (5.9) | 14.7 (14.0) | 13.3 (11.3) | <0.001 |
| ICU | | 834 (18.1%) | 343 (34.8%) | 433 (66.4%) | <0.001 |
| ICU LOS | | 1.1 (6.2) | 6.0 (13.0) | 9.1 (17.6) | <0.001 |
| ECMO | | 5 (0.1%) | 6 (0.6%) | 10 (1.5%) | <0.001 |
| Remdesivir | | 2,240 (49.0%) | 522 (53.8%) | 383 (59.1%) | <0.001 |
| Dexamethasone | | 2,273 (91.4%) | 517 (88.5%) | 428 (89.2%) | 0.046 |
| tocilizumab | | 162 (3.5%) | 68 (6.9%) | 77 (11.8%) | <0.001 |
| EI | | 4.0 (2.0–8.0) | 9.0 (7.0–12.0) | 8.0 (6.0–11.0) | <0.001 |
| Mortality | | 184 (4.0%) | 150 (15.2%) | 604 (92.6%) | <0.001 |

PAC: Post-Acute Care includes the intense rehabilitation facility, Skilled nursing facility and long-term acute care unit.

delta variants were BT, and 20.4% in alpha. The mean inpatient LOS was the highest in patients with delta variant (9.2 days SD: 11.5), and shortest in patients with the alpha (6.9 days, SD: 8.6) and omicron (6.9 days, SD: 8.4) variants. Increased PAC needs were associated with older age, and higher EI score. Higher albumin level, and breakthrough infection were

**Table 2. Descriptive Table of characteristics and demographics of COVID-19 inpatient based on variants.** Categorical variables expressed as count and percentage, and continuous variables as median and IQR.

| | | Total | Early | Alpha | Delta | Omicron | p-value |
|---|---|---|---|---|---|---|---|
| | | N = 6,248 | N = 4,729 (75%) | N = 1,105 (17.6%) | N = 245 (4.0%) | N = 169 (2.7%) | |
| Age | | 62.4 (44.7–75.6) | 64.6 (47.8–77.4) | 55.6 (38.5–67.5) | 57.5 (42.1–70.1) | 59.6 (38.9–74.8) | <0.001 |
| Race | White | 4,010 (67.2%) | 2,960 (65.5%) | 749 (71.6%) | 179 (74.6%) | 122 (73.9%) | <0.001 |
| | Black | 679 (11.4%) | 523 (11.6%) | 119 (11.4%) | 25 (10.4%) | 12 (7.3%) | |
| | Asian | 653 (10.9%) | 569 (12.6%) | 66 (6.3%) | 7 (2.9%) | 11 (6.7%) | |
| | Hispanic | 389 (6.5%) | 296 (6.6%) | 64 (6.1%) | 18 (7.5%) | 11 (6.7%) | |
| | Declined | 129 (2.2%) | 90 (2.0%) | 27 (2.6%) | 7 (2.9%) | 5 (3.0%) | |
| | Other | 107 (1.8%) | 78 (1.7%) | 21 (2.0%) | 4 (1.7%) | 4 (2.4%) | |
| Male | | 2,991 (47.9%) | 2,263 (47.9%) | 525 (47.5%) | 122 (49.8%) | 81 (47.9%) | 0.94 |
| BMI | | 29.5 (25.1–34.8) | 29.3 (25.0–34.4) | 30.5 (26.3–36.2) | 30.1 (25.8–37.5) | 28.5 (25.1–34.9) | <0.001 |
| Hemoglobin >15.7 g/dl | high | 184 (3.3%) | 137 (3.3%) | 39 (4.0%) | 4 (1.8%) | 4 (2.6%) | <0.001 |
| <11.7 g/dl | low | 2,141 (38.8%) | 1,657 (39.7%) | 336 (34.8%) | 72 (32.0%) | 76 (49.7%) | |
| 11.7–15.7 g/dl | normal | 3,196 (57.9%) | 2,383 (57.1%) | 591 (61.2%) | 149 (66.2%) | 73 (47.7%) | |
| WBC >11/ul | high | 764 (17.8%) | 515 (16.8%) | 220 (20.6%) | 12 (16.7%) | 17 (20.2%) | <0.001 |
| >4/ul | low | 297 (6.9%) | 161 (5.2%) | 118 (11.0%) | 8 (11.1%) | 10 (11.9%) | |
| 4-11/dl | normal | 3,230 (75.3%) | 2,391 (78.0%) | 730 (68.4%) | 52 (72.2%) | 57 (67.9%) | |
| Platelets >450/dl | high | 98 (1.8%) | 69 (1.7%) | 19 (2.0%) | 8 (3.6%) | 2 (1.4%) | 0.36 |
| <150/dl | low | 1,324 (24.6%) | 1,016 (24.9%) | 227 (24.1%) | 51 (22.9%) | 30 (20.4%) | |
| 150–450 /ul | normal | 3,970 (73.6%) | 2,997 (73.4%) | 694 (73.8%) | 164 (73.5%) | 115 (78.2%) | |
| Albumin | | 2.8 (2.5–3.1) | 2.8 (2.5–3.1) | 2.8 (2.5–3.1) | 2.7 (2.4–2.9) | 2.8 (2.5–3.1) | <0.001 |
| Blood type | O | 1,591 (40.5%) | 1,228 (40.4%) | 258 (38.8%) | 67 (46.9%) | 38 (46.9%) | 0.23 |
| | A | 1,545 (39.3%) | 1,179 (38.8%) | 283 (42.6%) | 56 (39.2%) | 27 (33.3%) | |
| | B | 597 (15.2%) | 477 (15.7%) | 93 (14.0%) | 13 (9.1%) | 14 (17.3%) | |
| | AB | 196 (5.0%) | 156 (5.1%) | 31 (4.7%) | 7 (4.9%) | 2 (2.5%) | |
| Rh pos | | 3,514 (89.4%) | 2,733 (89.9%) | 588 (88.4%) | 122 (85.3%) | 71 (87.7%) | 0.23 |
| CRP | | 72.4 (32.7–128.0) | 73.0 (32.4–130.0) | 70.3 (33.0–119.0) | 76.5 (42.3–117.5) | 69.8 (19.0–130.0) | 0.35 |
| TNF | | 24.3 (18.8–34.8) | 24.0 (18.8–34.3) | 29.9 (20.9–47.0) | 28.1 (28.1–28.1) | | 0.63 |
| DDIMER 48h | | 1.1 (0.6–2.1) | 1.1 (0.6–2.2) | 1.0 (0.6–1.9) | 1.1 (0.6–1.8) | 0.9 (0.6–2.3) | 0.46 |
| Troponin | | 1.4 (23.5) | 0.5 (4.9) | 0.8 (9.2) | 12.7 (58.4) | 52.8 (189.1) | <0.001 |
| Inpatient LOS | | 8.0 (9.1) | 8.2 (9.1) | 6.9 (8.6) | 9.2 (11.5) | 6.9 (8.4) | <0.001 |
| ICU | | 1,610 (25.8%) | 1,282 (27.1%) | 247 (22.4%) | 55 (22.4%) | 26 (15.4%) | <0.001 |
| ICU_Days | | 2.7 (9.8) | 2.8 (10.6) | 2.1 (6.1) | 3.3 (9.2) | 1.3 (6.2) | 0.021 |
| ECMO | | 21 (0.3%) | 13 (0.3%) | 7 (0.6%) | 0 (0.0%) | 1 (0.6%) | 0.20 |
| Remdesivir | | 3,145 (50.8%) | 2,264 (48.4%) | 660 (60.3%) | 155 (63.5%) | 66 (39.8%) | <0.001 |
| Dexamethasone | | 3,218 (90.6%) | 2,315 (89.9%) | 658 (93.7%) | 167 (97.1%) | 78 (77.2%) | <0.001 |
| tocilizumab | | 307 (4.9%) | 131 (2.8%) | 122 (11.0%) | 46 (18.8%) | 8 (4.7%) | <0.001 |
| EI | | 6.0 (3.0–9.0) | 6.0 (3.0–10.0) | 5.0 (2.0–8.0) | 5.0 (3.0–8.0) | 6.0 (3.0–10.0) | <0.001 |
| Breakthrough | | 455 (7.3%) | 66 (1.4%) | 225 (20.4%) | 70 (28.6%) | 94 (55.6%) | <0.001 |
| Mortality | | 938 (15.0%) | 796 (16.8%) | 110 (10.0%) | 22 (9.0%) | 10 (5.9%) | <0.001 |

associated with discharge home following hospitalization. See Table 2 for descriptive analysis of the discharge disposition based on variants.

## Baseline characteristics of vaccinated versus non-vaccinated: (see Table 3)

The median age of patients with breakthrough infections (67.5 years, IQR: 54.8–79.0) was older than that of those who were non-vaccinated (62.0 years, IQR: 44.1–75.2). Those with BT infection were less often admitted to the ICU (18%) compared to the non-vaccinated patients

**Table 3. Comparison of vaccinated and non-vaccinated SARS-CoV-2 patients admitted to hospital.** Categorical variables expressed as count and percentage, and continuous variables as median and IQR.

| | | Total | Non-Vaccinated | Vaccinated | p-value |
|---|---|---|---|---|---|
| | | N = 6,248 | N = 5,793 | N = 455 | |
| Age | | 62.4 (44.7–75.6) | 62.0 (44.1–75.2) | 67.5 (54.8–79.0) | <0.001 |
| Race | White | 4,010 (67.2%) | 3,660 (66.1%) | 350 (81.2%) | <0.001 |
| | Black | 679 (11.4%) | 650 (11.7%) | 29 (6.7%) | |
| | Asian | 653 (10.9%) | 632 (11.4%) | 21 (4.9%) | |
| | Hispanic | 389 (6.5%) | 372 (6.7%) | 17 (3.9%) | |
| | Declined | 129 (2.2%) | 119 (2.1%) | 10 (2.3%) | |
| | Other | 107 (1.8%) | 103 (1.9%) | 4 (0.9%) | |
| Male | | 2,991 (47.9%) | 2,756 (47.6%) | 235 (51.6%) | 0.094 |
| BMI | | 29.5 (25.1–34.8) | 29.5 (25.1–34.8) | 29.0 (24.8–34.2) | 0.41 |
| Hemoglobin >15.7 g/dl | high | 184 (3.3%) | 177 (3.5%) | 7 (1.7%) | <0.001 |
| <11.7 g/dl | low | 2,141 (38.8%) | 1,938 (38.0%) | 203 (48.4%) | |
| 11.7–15.7 g/dl | normal | 3,196 (57.9%) | 2,987 (58.5%) | 209 (49.9%) | |
| WBC >11/ul | high | 764 (17.8%) | 689 (17.6%) | 75 (19.9%) | 0.013 |
| >4/ul | low | 297 (6.9%) | 259 (6.6%) | 38 (10.1%) | |
| 4-11/ul | normal | 3,230 (75.3%) | 2,967 (75.8%) | 263 (69.9%) | |
| Platelets >450/ul | high | 98 (1.8%) | 90 (1.8%) | 8 (1.9%) | 0.94 |
| >150/ul | low | 1,324 (24.6%) | 1,220 (24.5%) | 104 (25.1%) | |
| 150-450/ul | normal | 3,970 (73.6%) | 3,668 (73.7%) | 302 (72.9%) | |
| Albumin | | 2.8 (2.5–3.1) | 2.8 (2.5–3.1) | 2.8 (2.5–3.1) | 0.75 |
| Blood type | O | 1,591 (40.5%) | 1,481 (40.5%) | 110 (40.1%) | 0.094 |
| | A | 1,545 (39.3%) | 1,422 (38.9%) | 123 (44.9%) | |
| | B | 597 (15.2%) | 567 (15.5%) | 30 (10.9%) | |
| | AB | 196 (5.0%) | 185 (5.1%) | 11 (4.0%) | |
| Rh pos | | 3,514 (89.4%) | 3,275 (89.6%) | 239 (87.2%) | 0.22 |
| CRP | | 72.4 (32.7–128.0) | 72.0 (33.0–128.0) | 74.5 (32.0–133.0) | 0.72 |
| TNF | | 24.3 (18.8–34.8) | 24.5 (18.8–34.8) | 20.9 (20.9–20.9) | 0.63 |
| D DIMER | | 1.1 (0.6–2.1) | 1.1 (0.6–2.1) | 1.0 (0.6–2.1) | 0.62 |
| Troponin | | 1.4 (23.5) | 1.3 (24.0) | 2.8 (9.6) | 0.47 |
| Inpatient LOS | | 8.0 (9.1) | 8.0 (9.2) | 7.1 (8.6) | 0.029 |
| No/% in ICU | | 1,610 (25.8%) | 1,528 (26.4%) | 82 (18.0%) | <0.001 |
| ICU Days | | 2.7 (9.8) | 2.8 (10.0) | 1.6 (6.0) | 0.016 |
| ECMO | | 21 (0.3%) | 20 (0.3%) | 1 (0.2%) | 0.66 |
| Remdesivir | | 3,145 (50.8%) | 2,933 (51.1%) | 212 (47.1%) | 0.10 |
| Dexamethasone | | 3,218 (90.6%) | 2,974 (90.7%) | 244 (90.4%) | 0.87 |
| Tocilizumab | | 307 (4.9%) | 281 (4.9%) | 26 (5.7%) | 0.41 |
| EI | | 6.0 (3.0–9.0) | 6.0 (3.0–9.0) | 7.0 (4.0–11.0) | <0.001 |
| Mortality | | 938 (15.0%) | 880 (15.2%) | 58 (12.7%) | 0.16 |

(26.4%, p<0.001). Those with BT had a higher median EI (7.0, IQR: 4.0–11.0), compared to those who were not vaccinated (6.0, IQR: 3.0–9.0) (see Table 3).

## Association of factors to discharge disposition

A multinomial model was constructed to explore the association between hospital discharge disposition and the following baseline factors known to impact outcomes from SARS-CoV-2 infection (see Table 4). We first performed a multinomial logistic regression analysis of

**Table 4.  Multinomial logistic regression of discharge dispositions i.e., post-acute care or died in hospital with the base outcome for comparison being home.**

| Post-acute care | | | | |
|---|---|---|---|---|
| | RRR | P value | 95% Confidence Interval | |
| Age | 1.048 | 0 | 1.04 | 1.05 |
| Variant | | | | |
| Alpha | 0.58 | 0 | 0.44 | 0.78 |
| Delta | 0.64 | 0.08 | 0.38 | 1.05 |
| Omicron | 0.63 | 0.13 | 0.34 | 1.15 |
| Breakthrough | 0.67 | 0.03 | 0.47 | 0.97 |
| Race | | | | |
| Black | 0.64 | 0.006 | 0.47 | 0.88 |
| Asian | 0.48 | 0 | 0.34 | 0.67 |
| Hispanic | 0.61 | 0.06 | 0.37 | 1.02 |
| BMI | 0.99 | 0.26 | 0.98 | 1 |
| Male | 0.94 | 0.51 | 0.79 | 1.12 |
| Dyspnea | 0.66 | 0 | 0.55 | 0.8 |
| Fatigue | 1.12 | 0.17 | 0.94 | 1.34 |
| Fever | 1.14 | 0.18 | 0.93 | 1.38 |
| Palpitations | 1.12 | 0.17 | 0.94 | 1.34 |
| Adm WBC | 1.01 | 0.06 | 0.99 | 1.02 |
| Hemoglobin | 1.04 | 0.112 | 0.99 | 1.09 |
| Platelet | 1 | 0.34 | 0.99 | 1 |
| Albumin | 0.55 | 0 | 0.43 | 0.69 |
| EI | 1.19 | 0 | 1.16 | 1.21 |
| Died in hospital | | | | |
| Age | 1.06 | 0 | 1.06 | 1.07 |
| Variant | | | | |
| Alpha | 0.87 | 0.4 | 0.64 | 1.19 |
| Delta | 0.84 | 0.56 | 0.47 | 1.49 |
| Omicron | 0.63 | 0.27 | 0.28 | 1.43 |
| Breakthrough | 0.42 | 0 | 0.26 | 0.68 |
| Race | | | | |
| Black | 1.03 | 0.83 | 0.73 | 1.47 |
| Asian | 1.31 | 0.08 | 0.96 | 1.79 |
| Hispanic | 1.14 | 0.6 | 0.68 | 1.91 |
| BMI | 0.99 | 0.19 | 0.97 | 1 |
| Male | 1.29 | 0.01 | 1.06 | 1.58 |
| Dyspnea | 1.37 | 0.009 | 1.08 | 1.75 |
| Fatigue | 0.82 | 0.06 | 0.66 | 1.01 |
| Fever | 0.95 | 0.7 | 0.77 | 1.18 |
| Palpitations | 1.17 | 0.12 | 0.95 | 1.45 |
| Adm WBC | 1.01 | 0.01 | 1 | 1.03 |
| Hemoglobin | 1.01 | 0.66 | 0.95 | 1.07 |
| Platelet | 0.99 | 0 | 0.99 | 0.99 |
| Albumin | 0.34 | 0 | 0.26 | 0.45 |
| EI | 1.09 | 0 | 1.07 | 1.12 |

complete cases (i.e., only patients with complete data) with the outcome variables being home (baseline category), PAC and DH. We included the following variables: age, gender, having a breakthrough(BT) infection, virus variant (as determined as the dominant type at the time of having a PCR positive test), race/ethnicity (White, Asian, Black, Hispanic), BMI, COVID-19 symptoms at the time of presentation (fatigue, shortness of breath, fever, and palpitations), the Elixhauser Comorbidity Index (EI) and laboratory studies including white blood cell count (WBC) count, D-Dimer, CRP, platelet count and albumin. For laboratory studies we used the first available value within three days of admission. Prior to multiple imputation of missing variables, the model was statistically valid with an F statistic < 0.0001. Imputations were performed for each of the variables included in the above model except for age, race, and BMI. Imputation was performed using the auxiliary variable that contained data for the minimum oxygen saturation for the first 24 hours. A Stata postestimation command to test goodness of fit for a multinomial logistic regression model showed no difference between the observed and expected frequencies within 10 groups. We then generated areas under the curve to determine the classification accuracy of the logistical regression model to generate multiclass ROC curves for classification accuracy based on multinomial logistic regression using yielding an AUROC of 0.789.

Older age (RRR: 1.04, 95% CI: 1.041–1.055), and higher EI (RRR:1.19, 95% CI: 1.17–1.22) were associated with a higher rate of discharge to PAC versus home. Similarly, older age (RRR:1.07, 1.06–1.07) and higher EI (RRR:1.09, 95% CI: 1.07–1.13) were associated with higher rates of DH versus home. Blacks (RRR: 0.64, 95% CI: 0.47–0.88), Asians (RRR: 0.48, 95% CI: 0.34–0.67) and Hispanics (RRR: 0.58, 95% CI: 0.35–0.97) were less likely to receive PAC, compared to white patients (see Table 4).

Among COVID-19 symptoms, dyspnea on presentation was significantly associated with less frequent discharge to PAC (RRR: 0.66, 95% CI: 0.55-.80) compared to home, but was not associated with more frequent DH (RRR: 1.2, 95% CI: 0.98–1.59). No other symptoms at presentation were significantly associated with discharge status (Table 5 if accepted, production will need this reference to link the reader to the Table).

Relative to early strains, having the alpha variant was associated with less frequent PAC discharge versus home (RRR 0.59 CI 0.44–0.78) but viral strain was not associated with the risk of DH compared to discharge home. The risk for discharge to PAC (RRR: 0.68, CI: 0.47–0.98) and DH (RRR 0.43, CI: 0.26–0.68) compared to home was lower for those participants presenting with BT infections.

For laboratory studies including the inflammatory markers, for every 1 mg/dl increase in albumin the relative risk of discharge to DH is reduced by 0.34 versus home (RRR: 0.34, CI: 0.26–0.45). Similarly, every unit increase in platelet count (RRR: 0.99, CI: 0.99–0.99) was associated with lower risks of 0.99 discharge to DH versus home. Conversely higher CRP (RRR:1.01, CI: 1.00–1.01) and coagulation marker D-Dimer (RRR: 1.07, CI: 1.04–1.10) were both associated with an increased risks of DH relative to going home. WBC was not significantly associated with higher rates of any discharge disposition. Higher hemoglobin (RRR: 1.04, CI: 1.00–1.10), D-Dimer (RRR: 1.03 CI: 1.00–1.06), and CRP (RRR: 1.00, CI:1.01–1.04) were associated with a higher risk of PAC discharge relative to home.

## Characteristics of patients comparing the COVID 19 infections to influenza

Compared to influenza patients, the COVID-19 patients had a higher median age (62.4 years, IQR: 44.7–75.6 versus 47.8 years, IQR: 31.3–68.7). The COVID-19 patients had a higher mean EI compared to influenza patients (6.0, IQR: 3.0–9.0 versus 0.0, IQR: 0.0–8.0), p = 0.001). The

**Table 5. Analysis of discharge disposition based on symptom at presentation.** Categorical variables expressed as percent.

| | Total | Home | IRF | SNF | LTACH | Hospice/Expired | p-value |
|---|---|---|---|---|---|---|---|
| | N = 6,248 | N = 4,611 | N = 125 | N = 801 | N = 59 | N = 652 | |
| General aches | 5.1% | 5.1% | 2.0% | 3.8% | 10.0% | 6.7% | 0.057 |
| Sore Throat | 3.4% | 3.5% | 2.0% | 3.6% | 2.5% | 2.7% | 0.79 |
| Rhinorrhea | 4.9% | 4.7% | 3.1% | 5.7% | 2.5% | 5.4% | 0.63 |
| Nausea/Vomiting | 9.6% | 9.1% | 5.1% | 12.9% | 7.5% | 10.5% | 0.011 |
| Diarrhea | 7.1% | 6.7% | 6.1% | 8.0% | 5.0% | 8.6% | 0.41 |
| Fatigue | 12.0% | 10.4% | 13.3% | 17.2% | 12.5% | 16.2% | <0.001 |
| Dyspnea | 14.9% | 13.5% | 12.2% | 17.4% | 12.5% | 22.5% | <0.001 |
| Cough | 15.8% | 14.5% | 19.4% | 19.6% | 12.5% | 19.3% | <0.001 |
| Joint pain | 3.7% | 3.2% | 6.1% | 5.4% | 2.5% | 5.0% | 0.013 |
| Chest Pain | 6.0% | 5.9% | 3.1% | 6.7% | 5.0% | 6.7% | 0.62 |
| Brain fog | 0.0% | 0.0% | 0.0% | 0.0% | 0.0% | 0.0% | 0.99 |
| Depression | 5.0% | 4.6% | 4.1% | 6.1% | 5.0% | 6.5% | 0.19 |
| Muscle pain | 3.8% | 3.9% | 2.0% | 1.9% | 10.0% | 5.6% | 0.002 |
| HA | 8.6% | 8.4% | 9.2% | 9.7% | 2.5% | 8.8% | 0.52 |
| Fever | 13.9% | 13.0% | 15.3% | 15.8% | 12.5% | 17.5% | 0.027 |
| Palpitation | 11.8% | 10.6% | 7.1% | 15.3% | 15.0% | 16.8% | <0.001 |
| Rash | 4.7% | 4.5% | 2.0% | 4.9% | 5.0% | 6.1% | 0.32 |
| Hair loss | 0.2% | 0.2% | 0.0% | 0.1% | 0.0% | 0.2% | 0.99 |
| Loss of smell/taste | 1.6% | 2.0% | 0.0% | 0.9% | 0.0% | 0.4% | 0.009 |
| Insomnia | 1.9% | 1.4% | 2.0% | 4.2% | 2.5% | 2.3% | <0.001 |
| Difficulty Thinking | 0.1% | 0.1% | 0.0% | 0.3% | 0.0% | 0.0% | 0.29 |
| Difficulty Memory | 1.1% | 0.4% | 0.0% | 4.4% | 0.0% | 2.5% | <0.001 |
| Anxiety | 3.4% | 3.4% | 2.0% | 3.9% | 2.5% | 2.9% | 0.79 |

mortality rate in the COVID-19 group was higher compared to influenza group (9.6% versus 0.8% respectively, p = <0.001). COVID-19 patients had a longer median hospital LOS (5.0, IQR 2.9–9.1 days versus 0.2 IQR: 0.1–1.7). In total, 25.8% of the COVID-19 patients were admitted to the ICU, compared to 3.4% of the influenza patients see Tables 6 and 7 (if accepted, production will need this reference to link the reader to the Table).

## Discussion

We sought to determine the relationship between baseline factors including inflammatory markers, and COVID-19 symptoms at presentation and discharge disposition, taking into consideration the variant and vaccine status. Additionally, we hoped to contextualize our findings by comparing the discharge dispositions of patients who had been hospitalized with a primary diagnosis of influenza to determine if there might be any differences in PAC utilization.

In this analysis of 6,248 patients, those of Black, or Asian race or being of Hispanic ethnicity were of increased likelihood to be discharged home than to a PAC when compared to White patients in our study. It has been recognized from previous studies that Blacks and Latinos have increased rates SARS-CoV-2 infection, severity of disease and mortality [20]. However, in our analysis, discharge to PAC was found to be less common for these cohorts. The observation that race is independently associated with discharge home warrants further study of the disparities in access to options in discharge planning. In our analysis, while Blacks and Asians were less likely to be discharged to PAC, they were not more likely to die or be placed in hospice. The data on the incidence and mortality of COVID-19 on minorities is limited but

**Table 6. Descriptive Table: Comparison of the SARS-CoV-2 to influenza patients admitted to hospital.** Categorical variables expressed as count and percent, and continuous variables as median and IQR.

| | | COVID Patients | Influenza Patients | p-value |
|---|---|---|---|---|
| | | N = 6,248 | N = 4,370 | |
| Age | | 62.4 (44.7–75.6) | 47.8 (31.3–68.7) | <0.001 |
| Race | White | 4,010 (67.2%) | 2,810 (64.3%) | <0.001 |
| | Black | 679 (11.4%) | 686 (15.7%) | |
| | Asian | 653 (10.9%) | 493 (11.3%) | |
| | Hispanic | 389 (6.5%) | 20 (0.5%) | |
| | Declined | 129 (2.2%) | 93 (2.1%) | |
| | Other | 107 (1.8%) | 266 (6.1%) | |
| Male | | 2,991 (47.9%) | 1,828 (41.8%) | <0.001 |
| BMI | | 29.5 (25.1–34.8) | 28.0 (24.1–33.2) | <0.001 |
| Hgb on admission | | 12.4 (11.0–13.8) | 13.3 (12.1–14.5) | <0.001 |
| WBC on admission | | 6.4 (4.4–9.1) | 7.0 (5.3–9.0) | <0.001 |
| Platelet Count on admission | | 196.0 (149.0–258.0) | 187.0 (150.0–231.0) | <0.001 |
| Pre-admit Albumin | | 3.4 (2.8–3.8) | 3.7 (3.4–4.1) | <0.001 |
| Inpatient LOS | | 5.0 (2.9–9.1) | 0.2 (0.1–1.7) | <0.001 |
| ICU | | 1,610 (25.8%) | 149 (3.4%) | <0.001 |
| COPD | | 1,051 (16.9%) | 348 (12.6%) | <0.001 |
| CVD | | 1,122 (18.1%) | 311 (11.3%) | <0.001 |
| Heart Txp | | 24 (0.4%) | 4 (0.1%) | 0.058 |
| Kidney Txp | | 134 (2.2%) | 22 (0.8%) | <0.001 |
| Liver Txp | | 40 (0.6%) | 12 (0.4%) | 0.23 |
| Lung Txp | | 25 (0.4%) | 2 (0.1%) | 0.008 |
| Sleep Apnea | | 1,331 (21.4%) | 352 (12.7%) | <0.001 |
| T2DM | | 2,194 (35.3%) | 545 (19.7%) | <0.001 |
| Hypertension | | 4,121 (66.3%) | 1,240 (44.9%) | <0.001 |
| HFpEF | | 766 (12.3%) | 196 (7.1%) | <0.001 |
| CAD | | 1,540 (24.8%) | 408 (14.8%) | <0.001 |
| Any Liver Disease | | 1,049 (16.9%) | 233 (8.4%) | <0.001 |
| EI | | 6.0 (3.0–9.0) | 0.0 (0.0–8.0) | <0.001 |
| Mortality | | 600 (9.6%) | 36 (0.8%) | <0.001 |

BMI: Body mass index, Hgb: Hemoglobin, WBC: White count cell, LOS: Length of stay, ICU: Intensive care unit, COPD: Chronic obstructive pulmonary disease, CVD: Cerebrovascular disease, TXP; transplantation, T2DM: Type 2 diabetes mellites, HFpEF: Heart failure with preserved ejection fraction, CAD: Coronary artery disease, EI: Elixhauser comorbidity score.

**Table 7. Discharge dispositions of SARS-CoV-2 and influenza inpatient populations.**

| | | SARS-COV-2 Patients | Influenza Patients | p-value |
|---|---|---|---|---|
| | | N = 5,593 | N = 4,370 | |
| Discharge disposition | Home | 4,098 (73.3%) | 4,085 (93.5%) | <0.001 |
| | Post-Acute Care | 905 (16.2%) | 258 (5.9%) | |
| | Hospice/Expired | 590 (10.5%) | 25 (0.6%) | |

expanding; however, in one report minorities, specifically Blacks in New York City, had a substantially higher mortality in the pandemic [21]. In general, Blacks were noted to have a disproportionate outcome burden in the pandemic [22]. Further studies are needed to understand the barriers to admission to rehabilitation facilities in the minorities. There have been previous studies in stroke populations which have shown that Blacks tend to have a less favorable outcomes if they survive when factoring in the impact of discharge disposition.

Several studies have highlighted the central role of inflammation in COVID-19 population in determining the disease severity and adverse clinical outcomes [23]; however, how these outcomes relate to discharge disposition, rehabilitation needs, and in and out of hospital mortality has not been studied. In this report, baseline CRP and D-Dimer were highest for those in the DH group, followed by PAC and home. In controlling for other factors, higher levels of these inflammatory markers on presentation are associated with dying in the hospital compared to discharge home. Not unexpectedly, patients discharged to PAC had a higher mortality rate compared to those discharged home based on the state death certificates. After controlling for CRP, WBC was not a significant predictor of disposition, but higher platelet counts were associated with lower rates of DH disposition. As newer therapies emerge for the management of SARS-CoV-2 infection, monitoring levels of these inflammatory markers may be a reasonable approach to predict outcomes.

While our primary focus was baseline factors that drive discharge disposition, we found that the patients who were discharged to PAC and DH had a longer inpatient LOS, were more likely to be placed into the ICU, and had a longer ICU stay during hospitalization. These findings were not unexpected considering that patients with a disposition PAC or DH had a higher EI. These values may be helpful in determining algorithms to in hospital care utilization for future surges of COVID-19.

Vaccination against SARS-CoV2 has been associated with mitigation of the severity of infection, and decreased rates of hospitalization [18]. In this analysis, presenting with a BT infection was associated with a lower risk of dying or utilizing PAC. Additionally, participants had a shorter length of hospitalization, were less likely to be transferred to the ICU, and when admitted into the ICU, they had a shorter ICU stay. This is an important consideration given the incomplete penetrance of vaccine uptake in the population. It will be important to assess over time the duration of protection offered by the initial vaccine and subsequent boosters. After controlling for vaccination status, we found that infection with the alpha variant was associated with being less likely to be discharged to PAC relative to home, compared to earlier strains.

A key strength of our study is the large sample size, which includes 62,279 PCR positive SARS-CoV-2 patients who presented to the healthcare system of whom 6248 patients who were admitted and were analyzed. This study offers a focused analysis of the discharge disposition and post-acute care needs of SARS-CoV-2 patients following hospitalization. Mutations and reinfections have been a challenge for the hospital capacity and understanding the PAC needs of the SARS-CoV-2 inpatients is an important factor for expense, planning and disparities. Another strength of the paper is the impact of vaccination on the discharge disposition and longer-term mortality following discharge further supporting planning and resource allocation during the pandemic.

One key limitation of the study is the inability to account for the factors inherent to the pandemic and local practices that determine the discharge disposition to the PAC during the pandemic. We have attempted to compare the SARS-CoV-2 inpatient population with a similar population of patients admitted to hospital with the primary diagnosis of influenza, outside the pandemic dates as a baseline for PAC admissions. The influenza patient database was chosen due to the similarities in disease spread and clinical burden when admitted and had a

similar hospital course, with need for ventilation secondary ARDS and effect on multiple organs. However, we recognize that the time course over which influenza patients were admitted was much longer and in a setting of markedly less constraint as experienced during the height of the pandemic and with the continued exodus of nursing staff from the work force. We did not think comparing the SARS-CoV-2 inpatient population to those traditionally admitted to the acute rehabilitation facilities with neurological deficits, traumatic brain injury or spinal cord injury would be a reasonable comparison. A recent study comparing the death rate of influenza and SARS-CoV-2 populations found that the decedents of SARS-CoV-2 were more likely to be male, between the age of 65 and 84 years, and were also more likely to have a higher incidence of a history of diabetes, hypertension, and obesity [24].

At a time of restricted resources and current crunch in labor in the pandemic, additional burden for hospitalization and PAC needs from the influenza patients in the flu season may add challenges to the healthcare system and needs further scrutiny and resource stratification. Appearance of additional variants of SARS-CoV-2 can also intensify the burden. Certainly resources i.e., staffing, and bed availability drive mortality [25], and understanding baseline predictors of patients including common symptoms of SARS-CoV-2 is of immense interest in the setting of constrained staffing models.

Long term morbidity following the prolonged hospitalization and the complications inherent to SARS- CoV-2 admission are not fully understood. While the functional and cognitive deficits that drive the discharges following hospitalization vary, the rehabilitation needs of these patients and a measure of return to baseline has not been quantified and remains ambiguous [26,27]. Furthermore, the impact on disability, quality of life (QoL) and workability remains to be studied. To the best of our knowledge, our study is the first to look at specific discharge dispositions of the SARS-COV-2 patients following hospitalization.

## Conclusion

In the setting of universally constrained healthcare staffing, understanding factors associated discharge disposition for patients admitted with COVID-19 remains of paramount importance. In this cohort of 6,248 patients admitted with SARS-CoV-2 infection, we observed that having a later variant and admission vaccination was associated with more favorable outcomes as indicated by discharge disposition. Further, older age, a greater burden of comorbidities, and longer hospital LOS, ICU LOS, and ICU admission are associated with discharge to PAC and dying in the hospital or discharge to hospice. The observation that race and ethnicity is independently associated with discharge home warrants further study of the disparities in discharge planning.

## Supporting information

**S1 Table. New onset SARS-COV-2 symptoms on presentation by discharge disposition using Fisher's exact test.**
(DOCX)

## Author Contributions

**Conceptualization:** Farha Ikramuddin, Leslie Morse.

**Data curation:** Farha Ikramuddin, Nguyen Nguyen.

**Formal analysis:** Farha Ikramuddin, Lianne Siegel.

**Methodology:** Farha Ikramuddin.

**Project administration:** Farha Ikramuddin.

**Supervision:** Leslie Morse.

**Validation:** Farha Ikramuddin.

**Visualization:** Farha Ikramuddin.

**Writing – original draft:** Farha Ikramuddin.

**Writing – review & editing:** Tanya Melnik, Nicholas E. Ingraham, Michael G. Usher, Christopher J. Tignanelli, Leslie Morse.

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
