## [Decision Letter · Decision Letter 0]

7 Apr 2022

PONE-D-21-40001Differences in Discharge Disposition following Hospitalizations for SARS-CoV-2 and influenzaPLOS ONE

Dear Dr. Ikramuddin,

Thank you for submitting your manuscript to PLOS ONE. After careful consideration, we feel that it has merit but does not fully meet PLOS ONE’s publication criteria as it currently stands. Therefore, we invite you to submit a revised version of the manuscript that addresses the points raised during the review process.

ACADEMIC EDITOR:

We now have two reviews of your study. You will see that both reviewers raised a number of strengths but also very clearly described their concerns with this study. Please note that one of the reviewers raised important issues related to the circulating variant(s) and vaccine uptake at the time of your study and the need to better contextualize your results within this. As well, both reviewers raised important points regarding the analysis and the need for a clearer presentation - and rationale - for certain aspects. 

We look forward to receiving your revised manuscript.

Kind regards,

Andrea Gruneir

Academic Editor

PLOS ONE

Journal Requirements:

Reviewers' comments:

Reviewer's Responses to Questions

**Comments to the Author**

1. Is the manuscript technically sound, and do the data support the conclusions?

Reviewer #1: Partly

Reviewer #2: Partly

2. Has the statistical analysis been performed appropriately and rigorously? 

Reviewer #1: Yes

Reviewer #2: Yes

3. Have the authors made all data underlying the findings in their manuscript fully available?

Reviewer #1: No

Reviewer #2: Yes

4. Is the manuscript presented in an intelligible fashion and written in standard English?

Reviewer #1: Yes

Reviewer #2: Yes

5. Review Comments to the Author

Reviewer #1: In this multicenter retrospective study, the authors describe and compare discharge dispositions following hospitalization among patients infected with SARS-CoV-2 and influenza. The authors find significant differences in the number of patients discharged to post-acute care (PAC) with a higher proportion of SARS-CoV-2 patients discharged to PAC, as well as longer length of hospitalization and higher mortality among patients with SARS-CoV-2. The authors also investigate associations between various clinical and laboratory parameters and discharge to PAC for patients with SARS-CoV-2. The strengths of the study include the large sample size as well as linking of patient data to state death certificates to characterize out of hospital death. My comments regarding this study are summarized below:

Major Comments:

1. Can the authors justify why these laboratory parameters were chosen to be included? Some of the variables such as CRP and albumin are more intuitive as markers of inflammation and nutritional status. Others are less so. For instance, why are blood type and Rh positive include in the analysis?

2. It is challenging to assess the clinical relevance of the differences found in laboratory parameters for hemoglobin, white blood cell count, platelets, etc. (Table 3). Most of the reported medians and interquartile ranges fall within the normal range of values for these variables. The authors could consider presenting these as categorical variables with cut-offs for low, normal, and high values. As it stands, it seems difficult to conclude that differences in WBC, where the medians range from 6 – 8, suggest significantly different states of inflammation.

3. For the multinomial logistic regression, how were variables of interest selected for inclusion in the model? Were these selected a-priori?

4. The rationale for evaluating presenting symptoms and association with discharge disposition is not entirely clear. Do you hypothesize that presenting symptoms may reflect severity of disease at presentation or potential extrapulmonary involvement? For instance, fever or body aches may suggest a more systemic inflammatory response.

5. The authors note that fewer Black and Asian patients are discharged to PACs compared to home. Is there literature regarding racial / ethnic disparities in hospital discharge disposition accounting for severity of illness and co-morbidities? This finding is worth noting in the discussion.

6. Is data available regarding where patients are coming from on hospital admission? (E.g. residing at home vs. SNF). Certainly, early in the pandemic many SNFs had large COVID outbreaks. This may impact the interpretation of results in the sense that some patients may be returning to SNF on discharge.

7. What is the minimum duration of observation between hospital discharge and out-patient death? In other words, were death certificates updated 3 months after the last discharge, 6 months etc.?

Minor Comments:

1. In the introduction, the significance is framed around the current influenza season. Given that the influenza season typically peaks in February or March in the US, the authors should consider updating some of the phrasing here.

2. Why does available data stop in November 2020 for influenza patients? As opposed to June 2021 for SARS-CoV-2 patients.

3. Page 11-12: It appears that there is duplicated text in the methods for the paragraphs starting with “State death certificates were linked”.

4. For clarity throughout the paper, I would suggest specifying that expired refers to in-hospital death throughout the text. Similarly, would clarify what “Mortality” in Table 2 and 3 refers to. Is this combined in and out of hospital death?

5. Table 3 - is there an error in median ICU LOS? Should this be presented as median (IQR) of ICU LOS for patients who were admitted to the ICU in each respective category.

6. Table 3 – There is no p-value for CRP.

7. Please include in the supplement which explanatory variables were excluded due to missingness and which were imputed. Missingness of 44% is quite high.

8. This may be a technical issue, but several of the mentioned supplemental tables and flow chart are not available on my review.

9. There are a few typos in the manuscript – E.g. Page 17 “between the age of 65 and 8418”.

Reviewer #2: Thank you for the opportunity to read your work.

The question asked seems relevant to me: what are the PAC needs for patients with Covid? Can we identify the predictive factors for final discharge to CAP?

Integrating patients with Influenza during other periods/years complicates the work and makes the results less readable. This question seems to me to belong to another study.

Introduction

The issue of SARS-Cov-2 variants is major. We know that the severity of the disease is strongly linked to the identified variant and to the vaccination status of the patient. this should be discussed in the introduction. In your work the variants have not been identified and it seems to me that this may be a major weakness in the work. You should specify the distribution of variants during the study period, and the vaccination rates of the general population and the population at risk of developing severe forms.

Methods

Discharge to PAC is very variable and depends on local practices, the availability of places in these structures, modes of financing, a medical and probably also social culture on the place of the elderly, and the possibilities of maintaining home for these patients.

I think it would have been interesting to describe the usual population (even admitted during the same period of study) admitted to your centers, those transferred to PAC, and describe the predictive factors of transfer.

The choice of influenza, as I can understand, is skewed if you haven't performed routine influenza screening in recent patients admitted for respiratory manifestations.

And the practices and the availability of places and PAC have certainly changed during the period of the study.

A flowchart of the patients included in the study should be presented. The tested, the admitted with SARS-Cov-2 and Influenza virus.

The severity of patients, their previous condition, and their condition at discharge are also variables to be assessed. Differences between centers and between age categories, etc., may also influence the final decision. This is not discussed in your introduction nor in the method.

What means hospice/expired? It is a composite end-point?

Results

I have a little trouble understanding the care pathways: how many came from outpatient facilities and how many were diagnosed and probably immediately serious in the emergency room?

I think we need to reduce the text and better present the results in tables. The flu vs Covid comparisons make it difficult to understand the results.

Patients with SARS Cov 2 are more serious, more often polypathological, and older. Their hospital stay is more favourable, shorter and require less sheave. Can we compare them on the orientations at the exit?

In the prediction model, did you include patients with Influenza? If so, the Covid does not appear as a predictive factor?

Or are it only the identified factors that explain the observed differences?

Discussion

The first paragraph should state the strengths of your work and your main results, your most important messages.

The discussion on mortality (in the general population and in-hospital...), the clinical characteristics and the severity factors of Influenza make it difficult to understand the text.

I don't understand the third paragraph, due to the discussion of PAC needs of flu patients.

The fourth paragraph: I don't think you can really conclude that the bed needs of patients with SARS Cov 2 are higher than those of patients with Influenza. The context and overload of the healthcare system was not the same during the pandemic period as in previous years during periods of seasonal flu.

Logic would have wanted to discuss first the predictive value of CAP of the variables related to the terrain, then the clinical picture, then the biological parameters and clinical characteristics, then the variables related to severity then the variables related to the hospital stay.

A question: the transfer rates were the same in the 12 centers? Did you notice a center effect? This can be important in the discussion of center practices and habits.

You discuss inflammatory markers in Covid, but you haven't assessed C-reative protein, a recognized severity marker in those patients. Could you be more precise?

Conclusion

I am not sure that you can say that the PAC needs of patients admitted for SARS-Cov-2 are higher than those of patients admitted for Influenza.

I don't understand why you say the predictors of discharge to PAC are different...or the presentation in the text and tables is confusing.

Tables

Tables 1 and 2

Table 1 too short with too long presentation of results in Results. Table 1 and 2 should be merged and simplified.

Table 3

These results are very briefly discussed in the discussion. Is it really important to keep them?

Chart 4

This analysis only concerns patients with SARS-Cov-2 or does it include patients with Influenza?

I don't see Covid/influenza in the variables.

6. PLOS authors have the option to publish the peer review history of their article (what does this mean?). If published, this will include your full peer review and any attached files.

Reviewer #1: No

Reviewer #2: No

---

## [Author Response · Author response to Decision Letter 0]

2 Dec 2022

Thank you for the constructive and important comments made, we are resubmitting the manuscript after major revision. As per your comments, added variants and vaccination status to the analysis.

---

## [Decision Letter · Decision Letter 1]

22 Dec 2022

PONE-D-21-40001R1Predictors of Discharge Disposition and Mortality following Hospitalization with SARS-CoV-2 InfectionPLOS ONE

Dear Dr. IKRAMUDDIN,

Thank you for submitting your manuscript to PLOS ONE. After careful consideration, we feel that it has merit but does not fully meet PLOS ONE’s publication criteria as it currently stands. Therefore, we invite you to submit a revised version of the manuscript that addresses the points raised during the review process.

Please revise.

We look forward to receiving your revised manuscript.

Kind regards,

Academic Editor

PLOS ONE

Journal Requirements:

Reviewers' comments:

Reviewer's Responses to Questions

**Comments to the Author**

1. If the authors have adequately addressed your comments raised in a previous round of review and you feel that this manuscript is now acceptable for publication, you may indicate that here to bypass the “Comments to the Author” section, enter your conflict of interest statement in the “Confidential to Editor” section, and submit your "Accept" recommendation.

Reviewer #3: (No Response)

Reviewer #4: (No Response)

2. Is the manuscript technically sound, and do the data support the conclusions?

Reviewer #3: Yes

Reviewer #4: Yes

3. Has the statistical analysis been performed appropriately and rigorously? 

Reviewer #3: Yes

Reviewer #4: I Don't Know

4. Have the authors made all data underlying the findings in their manuscript fully available?

Reviewer #3: Yes

Reviewer #4: Yes

5. Is the manuscript presented in an intelligible fashion and written in standard English?

Reviewer #3: Yes

Reviewer #4: Yes

6. Review Comments to the Author

Reviewer #3: This submission by Ikramuddin et al performed retrospective analysis on a cohort of 62,279 hospitalized SARS-CoV-2 positive patients. They investigated factors associated with either discharge home (-/+ Mortality), or to a inpatient post acute setting (-/+ death) and those that died in hospital.

They found that discharge to PAC and in hospital death was associated with older age, higher EI, along with elevated CRP and D-dimer levels (factors associated with worse outcome in SARS-CoV-2 infection) compared to those that were discharged home.

I was invited as an additional reviewer and I was unable to see the tracked changes version of the manuscript, so I cannot comment on how the manuscript has changed since the previous version.

I may have missed something, but cannot see any mention of influenza patients and discharge to PAC or home, given that in the discussion the authors state that they wanted to investigate differences in PAC utilization.

The authors state that D-dimer is an inflammatory marker. D-dimer is a marker of coagulation activation and not inflammation, this needs to be clarified in the text.

Table 2 + 3 need to define cut-off values for "high, low and normal" for haemoglobin, WBC and plts.

Include units within the tables.

Reviewer #4: It is a very well written article. I do not have much comments in terms of improving the quality of the article.

7. PLOS authors have the option to publish the peer review history of their article (what does this mean?). If published, this will include your full peer review and any attached files.

Reviewer #3: No

Reviewer #4: No

---

## [Author Response · Author response to Decision Letter 1]

18 Jan 2023

PLEASE SEE ATTACHED RESPONSE TO THE REVIEWERS LETTER

---

## [Decision Letter · Decision Letter 2]

7 Feb 2023

PONE-D-21-40001R2Predictors of Discharge Disposition and Mortality following Hospitalization with SARS-CoV-2 InfectionPLOS ONE

Dear Dr. IKRAMUDDIN,

Thank you for submitting your manuscript to PLOS ONE. After careful consideration, we feel that it has merit but does not fully meet PLOS ONE’s publication criteria as it currently stands. Therefore, we invite you to submit a revised version of the manuscript that addresses the points raised during the review process.

Please revise.

We look forward to receiving your revised manuscript.

Kind regards,

Academic Editor

PLOS ONE

Journal Requirements:

Reviewers' comments:

Reviewer's Responses to Questions

**Comments to the Author**

1. If the authors have adequately addressed your comments raised in a previous round of review and you feel that this manuscript is now acceptable for publication, you may indicate that here to bypass the “Comments to the Author” section, enter your conflict of interest statement in the “Confidential to Editor” section, and submit your "Accept" recommendation.

Reviewer #3: (No Response)

Reviewer #4: All comments have been addressed

2. Is the manuscript technically sound, and do the data support the conclusions?

Reviewer #3: Yes

Reviewer #4: Yes

3. Has the statistical analysis been performed appropriately and rigorously? 

Reviewer #3: Yes

Reviewer #4: I Don't Know

4. Have the authors made all data underlying the findings in their manuscript fully available?

Reviewer #3: Yes

Reviewer #4: Yes

5. Is the manuscript presented in an intelligible fashion and written in standard English?

Reviewer #3: Yes

Reviewer #4: Yes

6. Review Comments to the Author

Reviewer #3: I thank the authors for addressing my comments and the manuscript is much improved.

Minor point

Could the authors please standardize how they state “d-dimer”. It changes throughout the manuscript; “D-dimer, d-dimer, D dimer, D-DIMER, DDIMER”.

Reviewer #4: It is a very well written article. I do not have much comments in terms of improving the quality of the article.

7. PLOS authors have the option to publish the peer review history of their article (what does this mean?). If published, this will include your full peer review and any attached files.

Reviewer #3: No

Reviewer #4: No

---

## [Author Response · Author response to Decision Letter 2]

11 Feb 2023

The format of "D-Dimer" has been made uniform throughout the paper and is now based on NIH format. 

Thank you

---

## [Decision Letter · Decision Letter 3]

7 Mar 2023

Predictors of Discharge Disposition and Mortality following Hospitalization with SARS-CoV-2 Infection

PONE-D-21-40001R3

Dear Dr. IKRAMUDDIN,

We’re pleased to inform you that your manuscript has been judged scientifically suitable for publication and will be formally accepted for publication once it meets all outstanding technical requirements.

Kind regards,

Academic Editor

PLOS ONE

Additional Editor Comments (optional):

Reviewers' comments:

Reviewer's Responses to Questions

**Comments to the Author**

1. If the authors have adequately addressed your comments raised in a previous round of review and you feel that this manuscript is now acceptable for publication, you may indicate that here to bypass the “Comments to the Author” section, enter your conflict of interest statement in the “Confidential to Editor” section, and submit your "Accept" recommendation.

Reviewer #3: All comments have been addressed

Reviewer #4: All comments have been addressed

2. Is the manuscript technically sound, and do the data support the conclusions?

Reviewer #3: Yes

Reviewer #4: Yes

3. Has the statistical analysis been performed appropriately and rigorously? 

Reviewer #3: Yes

Reviewer #4: I Don't Know

4. Have the authors made all data underlying the findings in their manuscript fully available?

Reviewer #3: Yes

Reviewer #4: Yes

5. Is the manuscript presented in an intelligible fashion and written in standard English?

Reviewer #3: Yes

Reviewer #4: Yes

6. Review Comments to the Author

Reviewer #3: (No Response)

Reviewer #4: It is a very well written article. I do not have much comments in terms of improving the quality of the article.

7. PLOS authors have the option to publish the peer review history of their article (what does this mean?). If published, this will include your full peer review and any attached files.

Reviewer #3: No

Reviewer #4: No

---

## [Editor Report · Acceptance letter]

3 Apr 2023

PONE-D-21-40001R3 

Predictors of Discharge Disposition and Mortality following Hospitalization with SARS-CoV-2 Infection 

Dear Dr. Ikramuddin:

I'm pleased to inform you that your manuscript has been deemed suitable for publication in PLOS ONE. Congratulations! Your manuscript is now with our production department. 

Kind regards, 

on behalf of

Dr. Robert Jeenchen Chen 

Academic Editor

PLOS ONE